# Adenylsuccinate Synthetase MoADE12 Plays Important Roles in the Development and Pathogenicity of the Rice Blast Fungus

**DOI:** 10.3390/jof8080780

**Published:** 2022-07-26

**Authors:** Zhen Zhang, Zhongna Hao, Rongyao Chai, Haiping Qiu, Yanli Wang, Jiaoyu Wang, Guochang Sun

**Affiliations:** State Key Laboratory for Managing Biotic and Chemical Threats to the Quality and Safety of Agro-Products, Institute of Plant Protection and Microbiology, Zhejiang Academy of Agricultural Sciences, Hangzhou 310021, China; zhangz@zaas.ac.cn (Z.Z.); haozn@zaas.ac.cn (Z.H.); chairy@zaas.ac.cn (R.C.); qiuhp@zaas.ac.cn (H.Q.); wangyl@zaas.ac.cn (Y.W.)

**Keywords:** *Magnaporthe oryzae*, adenylsuccinate synthetase, asexual development, pathogenicity

## Abstract

Purines are basic components of nucleotides in living organisms. In this study, we identified the ortholog of adenylosuccinate synthase *MoADE12* in *Magnaporthe oryzae* by screening for growth-defective T-DNA insertional mutants. Gene replacement was performed to investigate the biological role of *MoADE12*. *Δmoade12* mutants were adenine auxotrophs that failed to produce conidia, and showed reduced perithecia formation and pathogenicity. Moreover, the *Δmoade12* mutant was hypersensitive to Congo red and oxidants, indicating that *MoADE12* was required for cell wall integrity and oxidative stress resistance. Transcriptomic analysis identified the underlying mechanisms and indicated that several pathogenicity-related genes were regulated in the *Δmoade12* mutant. Therefore, our data suggest that the adenylosuccinate synthase *MoADE12* is involved in the de novo AMP biosynthesis pathway and is important for conidiation and pathogenicity in the rice blast fungus.

## 1. Introduction

*Magnaporthe oryzae* is a hemibiotrophic fungal pathogen that causes rice blast disease, one of the most devastating rice diseases worldwide [1,2]. *M. oryzae* infection is initiated by the attachment of a three-celled conidium on the host surface. After germination and appressorial maturation on the surface of the plant, the penetration peg forms under the appressorium and penetrates the plant cuticle under high turgor pressure [3,4]. Endogenous resources in conidia, including glycogen, trehalose, polyols, and lipid bodies, are sufficient to support the development of infection structures in this process [5]. Once the fungus has penetrated into the first epidermal cell, the development of invasive hyphae requires the uptake of nutrients from the infected host site. During compatible interactions between *M. oryzae* and the plant host, the fungus derives its nutrients from living plant cells by conserved sensitive global genetic regulatory mechanisms of carbon and nitrogen metabolism before switching to the necrotrophic stage [6,7]. These regulatory pathways, including the target of rapamycin (TOR) signaling cascade, carbon catabolite repression (CCR), and nitrogen metabolite repression (NMR), confer the ability to utilize a wide range of carbon and nitrogen sources, which is conserved in fungi and enable them to colonize diverse ecological niches, including infected host cells [6,8]. In *M. oryzae*, the metabolic regulation of carbon and nitrogen, as well as their roles in growth, development, and pathogenicity have been well described [7,9,10,11]. In addition to the direct utilization of nutrients from host cells, many essential factors need to be synthesized in the pathogen cells, such as amino acids and vitamins [12]. In *M. oryzae*, several amino acid synthesis pathways, including those of arginine, lysine, leucine, and methionine, reportedly play important roles in development and pathogenicity [13,14,15,16]. 

Purines are the basic components of nucleotides in living organisms. In yeast, purines are synthesized via the de novo biosynthetic and salvage pathways [17]. De novo purine biosynthesis involves a ten-step reaction to convert 5-phospho-α-D-ribose 1-diphosphate (PRPP) into the first purine nucleotide, i.e., inosine monophosphate (IMP), which is the common precursor of both adenosine monophosphate (AMP) and guanosine monophosphate (GMP). Subsequently, IMP reacts with aspartate via adenylosuccinate synthetase (ADE12) to form adenylosuccinate, which is then converted to AMP by catalyzing adenylosuccinate lyase (ADE13). On the other hand, oxidation of IMP by xanthosine monophosphate (XMP), followed by amidation via GMP synthase (GUA1), results in the conversion of IMP to GMP. The purine salvage pathway ensures the recycling of purines formed by nucleotide degradation, which is essential for cellular purine homeostasis [17]. Recently, in *M. oryzae*, two enzymes involved in purine synthesis were identified [18,19]. *MoADE1* encodes an N-succinyl-5-aminoimidazole-4-carboxamide ribotide synthetase, which ligates aspartate with 5’-phosphoribosyl-4-carboxy-5-aminoimidazole (CAIR) to form 5-amino-4-imidazole-N-succinocarboxamide ribonucleotide (SAICAR), a key step in the conversion of PRPP into IMP. Disrupting *MoADE1* blocks the de novo synthesis of IMP. During plant development, the *Δmoade1* mutant is aborted, which may be attributed to the prevalence of purine nutritional deficiency [19]. Another gene, *MoGUK2*, which encodes guanylate kinase that catalyzes the conversion of GMP to guanosine diphosphate (GDP), is involved in conidial germination, appressorial formation, and infectious hyphae growth, resulting in attenuated virulence [18]. However, the *Δmoguk2* mutant formed normal conidiophores with decreased conidia, and its sexual development ability was abolished. The activities of *MoADE1* and *MoGUK2* indicate differences in the biological functions of related genes in purine metabolism. Therefore, to better understand the biological function of de novo purine synthesis in *M. oryzae*, further analysis of the synthesis of different purine types is required.

Adenylsuccinate synthetase (*ADE12*) catalyzes the first committed step in the conversion of IMP to AMP, which is the precursor of adenosine triphosphate (ATP) synthesis [20]. In *S. cerevisiae*, *ADE12* inactivation resulted in adenine auxotrophy [21]. In this study, we identified and inactivated the *ADE12* orthologous gene *MoADE12* in *M*. *oryzae*. MoADE12 is involved in the de novo adenosine biosynthesis pathway and is important for development and virulence in the rice blast fungus.

## 2. Materials and Methods

### 2.1. Fungal Strains and Culture Conditions

*M. oryzae* strain Guy11 and its derivatives were grown at 28 °C on complete medium (CM) plates, as described by Talbot et al. [22]. Mycelia from three-day-old cultures in liquid CM were harvested in order to isolate genomic DNA and total RNA. Conidia were obtained by harvesting nine-day-old cultures grown on CM plates. For vegetative growth, mycelial plugs (3 mm × 3 mm) were placed on CM or minimal medium (GMM) and cultured at 28 °C for six days. Each experiment was performed in triplicate and was repeated three times.

### 2.2. Identification of the Gene with T-DNA Insertion by Whole-Genome Resequencing

Sequencing and analysis were performed by OE Biotech Co., Ltd. (Shanghai, China). Libraries were constructed using the TruSeq Nano DNA LT Sample Preparation Kit (Illumina, San Diego, CA, USA). Briefly, genomic DNA was sheared into fragments of ~350 bp using S220 focused ultrasonicators (Covaris, Woburn, MA, USA). The adapters were ligated to the 3’-end of the sheared fragments. After PCR amplification and purification, the final libraries were sequenced on the Illumina HiSeq X Ten platform (Illumina Inc., San Diego, CA, USA) and 150 bp paired-end reads were generated. Raw reads were subjected to a quality check and filtered using Fastp (Version 0.19.5). Clean reads were aligned to the reference genome (GCF_000002495.2) using the Burrows Wheeler Aligner (BWA, version 0.7.12). After alignment, Picard (http://broadinstitute.github.io/picard/, accessed on 5 December 2019, Version 4.1.0.0) was used to mark duplicate reads, and SAMtools (version 1.4) was used to convert the alignment result format. GATK (version 4.1.0.0) was used to identify the InDel sites.

### 2.3. Targeted Gene Replacement and Complementation for MoADE12

A dual plasmid for gene knockout was generated using homologous recombination as previously described [23]. Briefly, the flanking fragments of *MoADE12* (~1 kb upstream and 1 kb downstream) were amplified from *M. oryzae* genomic DNA and inserted into the p1300-KO plasmid [24], which flanked the hygromycin phosphotransferase cassette (HPH). The resulting plasmid pKO-MoADE12 was confirmed via sequence analysis and was used to transform *M. oryzae* Guy11 by *Agrobacterium tumefaciens*-mediated transformation (AtMT). In order to complement the gene deletion mutant, the full-length *MoADE12* gene and its native promoter region (~1500 bp upstream from the start codon) were amplified by PCR and cloned into the p1300BAR (with glufosinate ammonium resistance) plasmid [24]. The final plasmid was sequenced and transformed into a *MoADE12* deletion mutant in order to obtain a full-length complemented transformant. The primers used in this study are listed in Appendix A.

### 2.4. Pathogenicity Assays

For the pathogenicity assays, about 100 four-week-old rice seedlings (*Oryza sativa* cv. CO-39) were infected with six milliliters of conidial suspensions prepared in 0.25% gelatin at a concentration of 5 × 10^4^ conidia mL^−1^ using an artist’s airbrush with high-pressure air. The inoculated plants were placed in a moist chamber at 25 °C for 24 h in the dark. One day after inoculation, rice seedlings were maintained in a moist chamber with a photoperiod of twelve hours under fluorescent light for an additional five days to enable the full development of disease symptoms. Disease severity was scaled to evaluate the virulence of the tested strains, as described previously [25]. The experiments were repeated at least three times in triplicate, yielding similar results.

### 2.5. Conidiation, Sexual Reproduction, and Analysis of Infection-Related Morphogenesis

For conidiation observation, mycelial plugs of the indicated strains were placed on glass slides in order to induce conidiation under continuous light for 24 h and observed under a light microscope. For sexual reproduction, plugs of the *Δmoade12* mutant, wild-type Guy11 (MAT1-2), and mating partner strain TH3 (MAT1-1) were point-inoculated 3 cm apart on oatmeal medium (OMA) plates and cultured at 20 °C with continuous light for 3–4 weeks. Mature perithecia were crushed and photographed, in order to examine the asci and ascospores.

In order to analyze infection-related morphogenesis, conidia suspensions were inoculated onto plastic coverslips (Thermo Fisher Scientific, Waltham, MA, USA) to examine conidial germination and appressorium formation. The appressorium turgor pressure was determined using a cytorrhysis assay [26]. Infectious hyphal growth was assayed in the epidermal cells of the leaf sheaths of rice (*Oryza sativa* cv. CO-39), as described previously [27].

### 2.6. Assays for Osmotic Stress, Cell Wall Integrity and Oxidative Stress

In order to test the sensitivity against osmotic regulators, cell-wall-disrupting agents, or oxidants, vegetative growth was assayed on CM plates with sodium chloride, sorbitol, calcofluor white (CFW, Sigma-Aldrich, St. Louis, MO, USA), Congo red (CR), rose Bengal, or H_2_O_2_ (30% (*w*/*w*) in H_2_O) at the different concentrations described in the Results at 28 °C. All reagents were purchased from Sinopharm Chemical Reagent Co. Ltd (Shanghai, China), except for CFW. Growth rates of all tested strains were determined by measuring the colony diameter of 6-day-old cultures. All experiments were repeated at least three times. The mean ±SD of growth rate was determined using SPSS Statistics 22 (IBM, Inc., Armonk, NY, USA). Asterisks indicate a statistically significant difference at *p* < 0.05.

### 2.7. Transcriptomic Analysis

Mycelia of the tested strains from three-day-old cultures in liquid CM were harvested. Total RNA was extracted using the TRIzol reagent according to the manufacturer’s protocol (Invitrogen, Carlsbad, CA, USA). The libraries were constructed using the TruSeq Stranded mRNA LT Sample Prep Kit (Illumina, San Diego, CA, USA), according to the manufacturer’s instructions. Transcriptome sequencing and analysis were performed by OE Biotech Co., Ltd. The libraries were sequenced on an Illumina HiSeq X Ten platform, and 150 bp paired-end reads were generated. Differential expression analysis was performed using the DESeq2 R package (2012). A *q*-value < 0.05 and a fold-change > 2 or <1/2 were set as the threshold for significantly differential expression.

## 3. Results

### 3.1. Identification of MoADE12 in M. oryzae

In order to identify new pathogenicity-related genes in *M. oryzae*, we screened for T-DNA insertion mutants. The mutant M76.8 exhibited impaired aerial mycelial growth. In order to determine the insertion site of T-DNA in M76.8, whole-genome resequencing was performed. Sequence analyses showed that the T-DNA insertion occurred 573 bp before the start codon of the MGG_17000 open reading frame (Appendix A), which was annotated as adenylosuccinate synthetase, and is homologous to ADE12 in *Saccharomyces cerevisiae*. This gene was assigned as *MoADE12*.

In order to confirm whether *MoADE12* deletion leads to the same phenotype as observed in the mutant M76.8, the gene deletion mutant *Δmoade12* was generated by replacing the *MoADE12* gene with a hygromycin phosphotransferase gene cassette. In addition, the complemented strain *Δmoade12/MoADE12* was constructed by reintroducing the wild-type *MoADE12* gene cassette, including the native promoter, into the mutants (Appendix A). Similarly with M76.8, the *Δmoade12* strain was unable to grow on a 1% glucose (wt/vol) containing GMM. The growth of the mutants was recovered by supplementation with adenine, but not by hypoxanthine, in the GMM medium (Figure 1A). The complemented strain showed amelioration of the growth defects. The process of purine salvage can generate AMP from adenine, or IMP from hypoxanthine; however, our results indicated that AMP synthesis from IMP is disabled in the mutants *Δmoade12* and M76.8, and the loss of *MoADE12* was associated with growth defects in M76.8.

The strains *Δmoade12,* M76.8, Guy11, and *Δmoade12/MoADE12* were inoculated onto CM and OMA plates and incubated at 28 °C for six days. The radical growth rates of *Δmoade12* and M76.8 were similar to those of Guy11 or *Δmoade12/MoADE12.* However, the mutants *Δmoade12* and M76.8 formed almost no aerial hyphae, as indicated by a small amount of light-colored growth on the plate (Figure 1B). Therefore, *MoADE12* plays a role in the aerial growth of rice blast fungi.

Next, we compared the growth of the tested strains on GMM plates containing different amounts of adenine. The growth rates of the *Δmoade12* mutant corresponded well with the adenine concentration. The *Δmoade12* mutant did not grow well with an adenine concentration lower than 35 µM. The results suggested that the amount of adenine in the adenine-deficient medium was associated with the growth of the *MoADE12* disruption mutants (Figure 1C).

### 3.2. MoADE12 Is Involved in Asexual Conidiation and Sexual Reproduction

In order to determine whether *MoADE12* participates in reproductive development, three strains, Guy11, *Δmoade12*, and *Δmoade12/MoADE12*, were inoculated on CM plates to promote asexual development. The results showed that the *Δmoade12* mutant hardly formed conidiophores compared to the wild type and complemented transformants (Figure 2A). However, conidiation of the mutant *Δmoade12* was restored by supplementing the CM medium with 1 mM adenine (data not shown).

For sexual development, the tested strains and mating partner strain TH3 were inoculated onto OMA plates and cultured at 20 °C for three weeks. The results showed that the *Δmoade12* mutant produced perithecia, and the ability of the *Δmoade12* mutant to produce perithecia was visibly reduced compared that of the wild type and complemented transformant (Figure 2B). However, mature perithecia and asci were observed under the microscope in the *Δmoade12* mutant (Figure 2B). These results above indicate that *MoADE12* is essential for asexual development, and plays a role in sexual reproduction in *M. oryzae*.

### 3.3. Deletion of MoADE12 Significantly Attenuates the Virulence of M. oryzae

Since the *Δmoade12* mutant showed substantially reduced formation of conidia on CM plates, the plates were supplemented with 1 mM adenine to harvest the conidia of strains Guy11, *Δmoade12*, and *Δmoade12/MoADE12*, and to perform pathogenicity tests. Seven days after inoculation of the 4-week-old rice seedlings, the *Δmoade12* mutant caused few small lesions on rice leaves compared to the numerous typical lesions caused by Guy11 (Figure 3A). The lesion number decreased 82.3% compared to the wild type, and the mutant only produced uniform dark brown pinpoint lesions without visible centers. The attenuated virulence of the *Δmoade12* mutant was complemented and fully recovered by the reintroduction of *MoADE12*. The results indicated that *MoADE12* is required for the full virulence of *M. oryzae*.

In order to clarify why the deletion of *MoADE12* resulted in severely attenuated virulence, the infection-related morphogenesis of the *Δmoade12* mutant was examined. First, conidial germination, appressorial formation, and appressorial turgor were observed on the inductive surfaces. The results showed that no significant difference was present between the *Δmoade12* mutant and the wild type (data not shown), indicating that loss of *MoADE12* does not impair the normal function of appressoria derived from the conidia formed on adenine-supplemented media. Next, infectious hyphae of the *Δmoade12* mutant were observed in the epidermal cells of the rice leaf sheath. At 40 h post-inoculation, wild-type Guy11 had elaborated bulbous infectious hyphae within host cells, whereas the infectious development of the *Δmoade12* mutant was severely impaired, as it had formed primary hyphae alone and had not branched (Figure 3B). The results suggest that *MoADE12* is essential for infectious hyphal growth in rice cells.

### 3.4. MoADE12 Is Required for Cell Wall Integrity, Osmotic Stress, and Oxidative Stress Resistance

In order to examine whether deletion of *MoADE12* results in defective responses to abiotic stress, strains were inoculated on CM plates containing sodium chloride, sorbitol, calcofluor white, Congo red, rose Bengal, and H_2_O_2_. At six days post-inoculation, the *Δmoade12* mutant exhibited 72.0% and 61.0% growth reduction in the presence of 1 M sodium chloride and 50 μg/mL CR, respectively, and these phenotypes were more sensitive compared with those of Guy11 (Figure 4). However, no significant differences in growth inhibition rates of the *Δmoade12* mutant relative to Guy11 were observed upon treatment with sorbitol (1 M) and CFW (50 μg/mL). For the same molar concentration, the osmotic pressure of the NaCl solution was higher than that of sorbitol. CFW and CR interfere with cell wall assembly by binding to chitin and β-1,3 glucan, respectively [28]. These factors may account for the differences in the phenotypes when using the osmotic and cell wall-disrupting agents. Thus, *MoADE12* is required for cell wall integrity and resistance to high osmotic stress.

Under oxidative stress conditions, the growth of the *Δmoade12* mutant was significantly inhibited in the presence of the tested oxidants, compared with that of Guy11 (Figure 5). The *Δmoade12* mutant showed an increase in inhibition rates between 4.7–67.0%, compared to Guy11 in the presence of rose Bengal and H_2_O_2_ stress. Notably, the *Δmoade12* mutant was non-viable on a medium containing 0.03% H_2_O_2_. The results indicate an essential role for *MoADE12* in the response to oxidative stress in *M. oryzae*.

### 3.5. Transcriptional Analysis for Identification of Differentially Expressed Genes (DEGs) in a Comparison of the ∆moade12 and Wild-Type Strains

In order to clarify the genetic basis for the phenotypic defects caused by *MoADE12* deletion, comparative transcriptomic analysis between the *∆moade12* mutant and wild-type Guy11 strains was performed during hyphal growth in liquid CM media. A total of 2541 DEGs were identified between *∆moade12* and wild-type Guy11 using the standard criterion of |log_2_FC| > 1 and a *q*-value < 0.05, including 1059 up-regulated and 1482 down-regulated genes (Appendix A). Among the DEGs, 26 pathogenicity-related genes were identified with |log_2_FC| values > 1.5, including 20 down-regulated and 6 up-regulated genes (Table 1). The genes *SLP1* [29], *SPD2* [30], *SLP2* [29], and *AVRPi9* [31], which suppress host immunity, showed significantly higher expression levels.

In order to identify the biological processes regulated by MoADE12, gene ontology (GO) term analysis and Kyoto Encyclopedia of Genes and Genomes (KEGG) pathway annotations were performed. Based on sequence homology, 673 genes were categorized into different functional groups, of which 35 (*p*-value < 0.05) are shown in Figure 6. The enriched genes included those for catalytic (126 genes), transporter (39), membrane activity (31), and signaling (18). A total of 2509 genes were mapped to the reference pathways, and the top 20 enriched pathways are shown in Figure 7. The pathways showing significant enrichment (*p*-value < 0.05) included tyrosine (19); beta-alanine (12); glycine, serine, and threonine metabolism (19); isoquinoline alkaloid biosynthesis (13); phenylalanine metabolism (11); vitamin b6 metabolism (6); tropane, piperidine, and pyridine alkaloid biosynthesis (6); tryptophan metabolism (13); sulfur metabolism (8); cysteine and methionine metabolism (14); taurine and hypotaurine metabolism (5); starch and sucrose metabolism (16); glycosphingolipid biosynthesis-globo and isoglobo series (4); valine, leucine, and isoleucine biosynthesis (7); histidine metabolism (7); alanine, aspartate, and glutamate metabolism (9); glycerolipid metabolism (10); non-homologous end-joining (4); nitrogen metabolism (6); and betalain biosynthesis (6).

## 4. Discussion

In this study, we used whole-genome resequencing to identify *ADE12* orthologs in *M. oryzae* using the T-DNA insertion mutant M76.8, which showed abnormal mycelium growth. Subsequently, we confirmed our findings using targeted gene replacement and complementation analyses. The *Δmoade12* mutant and M76.8 were unable to grow on GMM plates, and growth was restored by supplementing with adenine but not with hypoxanthine (Figure 1A). This result suggests that the AMP salvage pathway mediates the growth defects of the *MoADE12* deletion mutant, which is consistent with findings in yeast [20].

The growth rate of the *Δmoade12* mutant cultured on CM or OMA plates was comparable to that of the wild type, but the growth of aerial hyphae in the mutants was inhibited. Deletion of *MoADE12* causes severe defects in conidiation and reduces the number of perithecia, and the results differ from those reported for two purine synthetic mutants [18,19]. In the de novo purine biosynthesis, MoADE1 is the key enzyme in the synthesis of IMP, which is the precursor of AMP and GMP. MoGUK2 catalyzes the conversion of GMP to GDP, and MoADE12 catalyzes the first committed step in the conversion of IMP to AMP. AMP and GMP can produce a variety of derivatives, some of which may mediate opposing biological responses [52]. Phenotypic differences of their gene deletion mutants may be a result of differences in their products and derivatives. Transcriptomic data analysis showed significant down-regulation of several genes associated with asexual reproduction, including *COS1*, MGG_12958, MGG_04699, MGG_06898, and MGG_02246 [40]. *COS1* is essential for conidiophore development, and regulates the expressions of genes responsible for conidiation, which is consistent with the conidiation defects observed in the *Δmoade12* strain. Notably, two G-protein signaling proteins, *MGG1* encoding *Gγ* subunit [46], and *RGS7* encoding regulators of G-protein signaling protein [34] were down-regulated in the *Δmoade12* strain. In fungi, G-protein signaling regulates two downstream signaling branches, i.e., the cAMP-dependent protein kinase (PKA) and mitogen-activated protein kinase (MAPK) signaling pathways, in order to regulate a variety of cellular functions such as conidiation, mating, and pathogenicity [53,54]. MoADE12 is a key enzyme that converts IMP to AMP, whereas MoGuk2 converts GMP to GDP to synthesize GTP, and MoADE1 provides the IMP synthesis precursor. These factors or their derivatives are essential for the activation of the G-protein and cAMP-dependent signaling pathways.

Our results showed that the conidia of the *Δmoade12* mutant harvested from a CM plate supplemented with adenine could germinate normally and form functional appressoria on an inductive surface, suggesting that adenines stored in conidia are sufficient to complete the invasion of host cells. Additionally, the initial colonization process does not appear to require de novo AMP biosynthesis during the early infection stage. However, the infectious hyphae of the *Δmoade12* mutant were inhibited in host cells compared with those of the wild type, suggesting that de novo AMP biosynthesis is required for the growth of infectious hyphae. Although adenine is abundant in rice leaf cells [55], Fernandez et al. found that the *Δmoade1* mutant is limited to the acquisition of adenine from rice cells, despite up-regulated expressions of two putative purine transporters (putative uric acid-xanthine permease gene *UAP1* and putative purine-cytosine permease gene *FCY2*) [19]. Five putative purine transporter genes are found in the *M. oryzae* genome based on homology search with the purine transporters of *Aspergillus nidulans* [56] besides *UAP1* and *FCY2*, including MGG_06740, MGG_08279, and MGG_00158. *UAP1* is homologous to *UapA* and *UapC* of *A. nidulans*, MGG_06740, and MGG_08279 to *AzgA*, *FCY2* and MGG_00158 to *FcyB*. The purine transporters have the highly specific ability to transport purines [56]. In *A. nidulans*, AzgA and FcyB are identified to transport adenine, but UapA and UapC transport xanthine and uric acid, respectively. In the present study, only the expression of *UAP1* were found to be down-regulated in the *Δmoade12* mutant, and the expressions of *AzgA* and *FcyB* homologous genes were not perturbed. Our results showed that the growth rate of the *Δmoade12* mutant was comparable to that of the wild type on CM media, even though the mutant is not able to synthesize AMP de novo. In fact, AMP or adenine and their analogues should be present in CM media due to the presence of yeast extract and peptone in the medium formulation. These facts imply that it may involve diverse salvage pathways. In human erythrocytes, four different pathways of adenine salvage and twelve different pathways of adenosine salvage have been found [57]. According to the KEGG pathway, we found that several amino acid and carbohydrate metabolism pathways were enriched, which allow the *Δmoade12* mutant to synthesize the necessary AMP in CM medium for growth.

Furthermore, our results showed that the *MoADE12* deletion mutant was hypersensitive to oxidative stress. Reactive oxygen species (ROS) are scavenged by antioxidant systems, including catalase and various peroxidases. In the genome of *M. oryzae*, a total of 27 peroxidase genes had been revealed [58]. In the present study, transcriptions of 13 genes were perturbed by deletion of *MoADE12*. Among those, 10 genes were down-regulated, including *MoLIP2*, *CPXB*, *CATB*, *MoHPX3*, *MoHPX2*, *NOX1*, *MoLIP3*, *MoPRX1*, *NOX2*, and *MoAPX1*. According to Mir et al. [58], six of those genes, including *CPXB*, *CATB*, *NOX1*, *MoLIP3*, *NOX2*, and *MoAPX1* were up-regulated under H_2_O_2_ stress, and nine of those genes, except for *MoHPX3*, were induced during the early phase of host infection. Furthermore, *CATB* [33], *NOX1* [43], *NOX2* [43], and *MoPRX1* [58] were required for pathogenicity by balancing the intracellular level of ROS or the efficient removal of extracellular ROS in *M. oryzae*. Although deletion of *CXPB* does not affect pathogenicity, *CXPB* is involved in fungal defense against H_2_O_2_ accumulated in epidermal cells of rice during the early stage of infection [59]. Such evidence suggested that the surge in ROS at the infection site is inhibitory for the growth of infectious hyphae of the *Δmoade12* mutant.

In conclusion, we identified a T-DNA mutant of *M. oryzae* that exhibited impaired aerial mycelial growth and attenuated virulence. Genome resequencing of the mutant led to the identification of the adenylosuccinate synthase encoding gene *MoADE12*, which catalyzes the first step in the de novo synthesis of AMP from IMP. Loss of *MoADE12* leads to the abolition of asexual development as well as hypersensitivity to oxidative stress. *MoADE12* is involved in many biological processes, including carbohydrate metabolism, amino acid metabolism, signal transduction, and stress responses. The associated regulatory networks should be evaluated in future studies.

## Figures and Tables

**Figure 1 jof-08-00780-f001:**
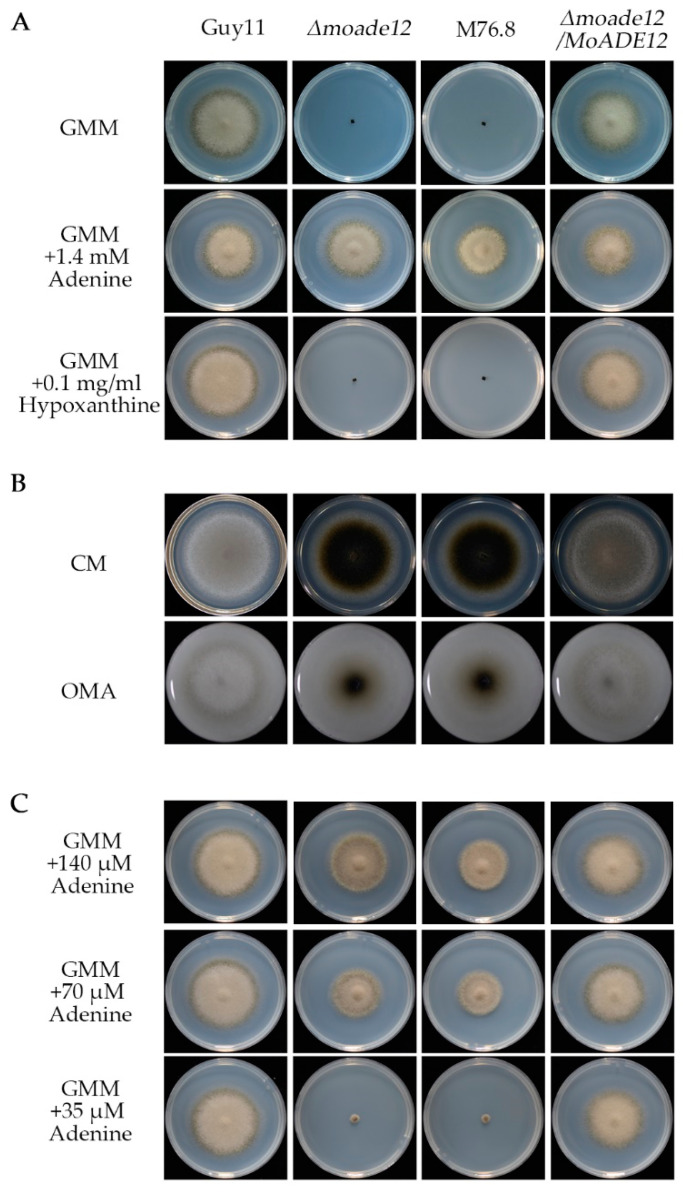
Deletion of the *MoADE12 gene* in *M. oryzae* resulted in adenine autotrophy, and reduced aerial hyphae growth on nutrient-rich media. (**A**) The *Δmoade12* mutant was grown on GMM plates (6-centimeter diameter) supplemented with adenine or hypoxanthine at the indicated concentrations for six days. (**B**) Colony morphology was examined on the CM and OMA plates (9-centimeter diameter) for nine days. (**C**) The *Δmoade12* mutant was grown on GMM plates (6-centimeter diameter) supplemented with adenine at the indicated concentrations for six days.

**Figure 2 jof-08-00780-f002:**
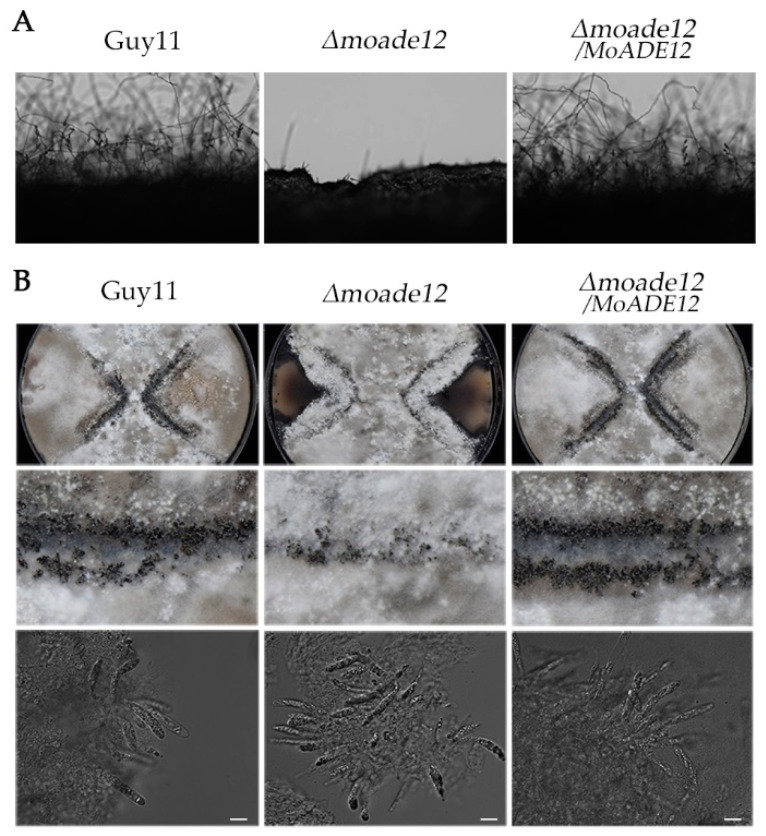
The *Δmoade12* mutant showed defects in asexual and sexual development. (**A**) Conidial development was examined under a light microscope at 20 h after induction on glass slides. (**B**) Perithecia (Rows 1 and 2) and asci (Row 3) formed by the indicated strains were photographed 3 weeks after inoculation. The white scale bar represents 20 µm.

**Figure 3 jof-08-00780-f003:**
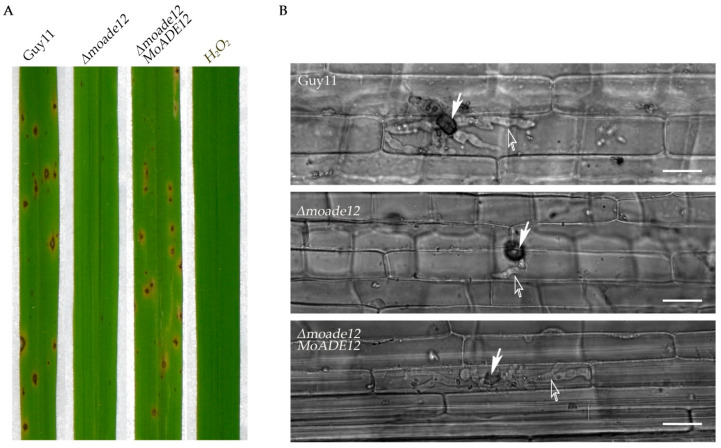
The *Δmoade12* mutant attenuates virulence and infectious hyphal observation. (**A**) Conidial suspensions from the indicated strains were sprayed onto four-week-old rice seedlings and photographed at seven days post-inoculation. (**B**) The *Δmoade12* mutant was defective in infectious hyphal growth. Appressorium is marked with the white solid arrow, and infectious hyphae is marked with the hollow white arrow. The white scale bar represents 20 µm.

**Figure 4 jof-08-00780-f004:**
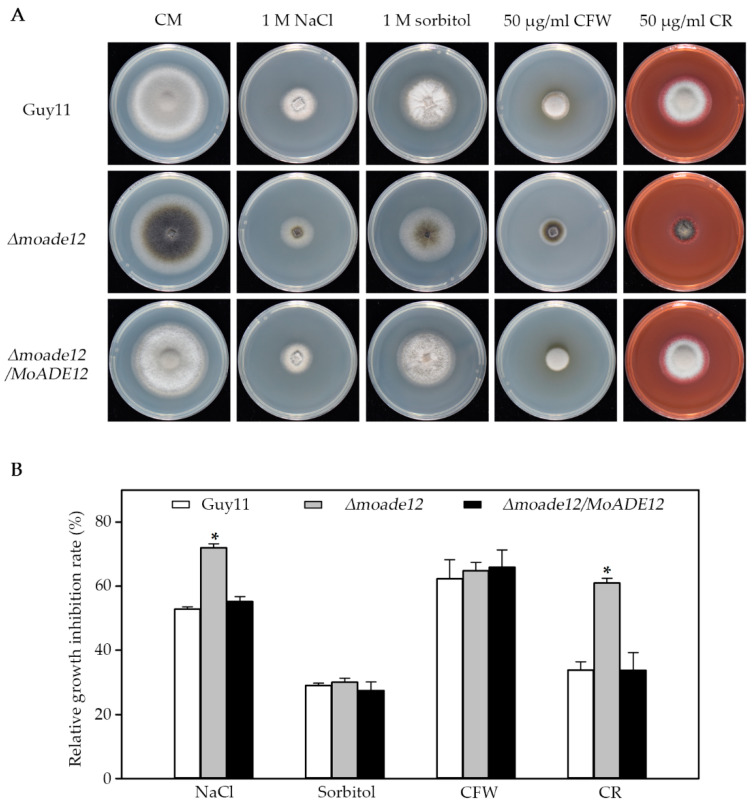
Effect of osmotic pressure and cell wall disruptors on vegetative growth of the indicated *M. oryzae* strains. (**A**) Vegetative growth of Guy11, *Δmoade12*, and *Δmoade12/MoADE12* were examined by growth on CM containing NaCl (1 M), sorbitol (1 M), CFW (50 μg/mL), and CR (50 μg/mL) plates at 28 °C for six days. (**B**) Growth inhibition rates of the indicated strains exposed to different chemicals. The experiments were performed in triplicate. ANOVA was performed after arcsine transformation of the growth inhibition rates. However, the original percentages are shown in the figure. Error bars represent the SD. Asterisks in each data column indicate significant differences at *p* < 0.05.

**Figure 5 jof-08-00780-f005:**
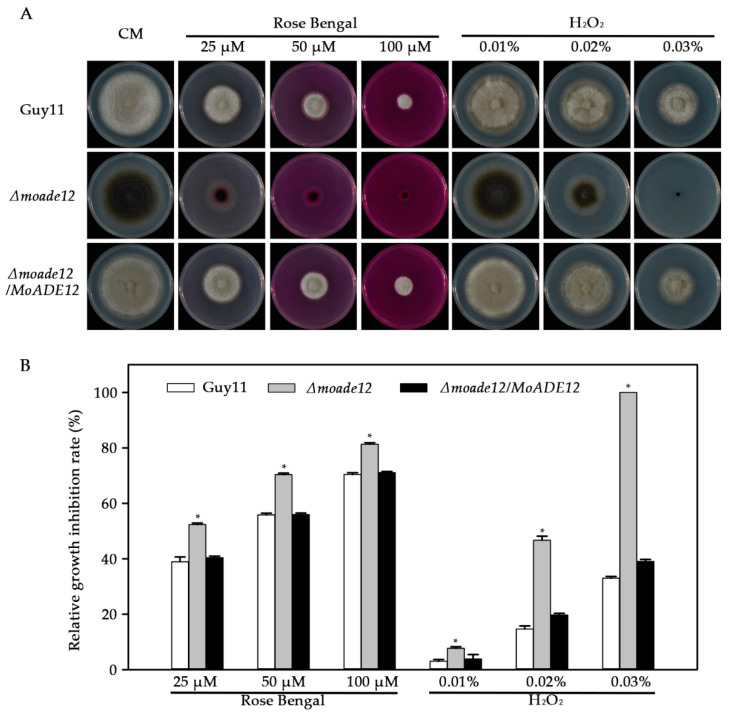
Effects of different oxidants on vegetative growth of the indicated *M. oryzae* strains. (**A**) Vegetative growth of Guy11, *Δmoade12*, and *Δmoade12/MoADE12* on CM containing rose Bengal (25 μM, 50 μM, and 100 μM) and H_2_O_2_ (0.01%, 0.02%, and 0.03% [*w*/*v*]) at 28 °C for six days. (**B**) Growth inhibition rates of the indicated strains administered different oxidants. The experiments were performed in triplicate. ANOVA was performed after arcsine transformation of the growth inhibition rates. However, the original percentages are shown in the figure. Error bars represent the SD. Asterisks in each data column indicate significant differences at *p* < 0.05.

**Figure 6 jof-08-00780-f006:**
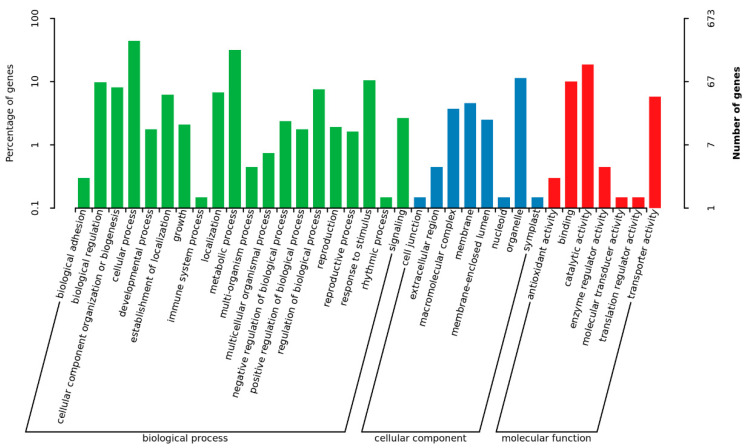
Functional grouping of differentially expressed genes in the *∆moade12* mutant.

**Figure 7 jof-08-00780-f007:**
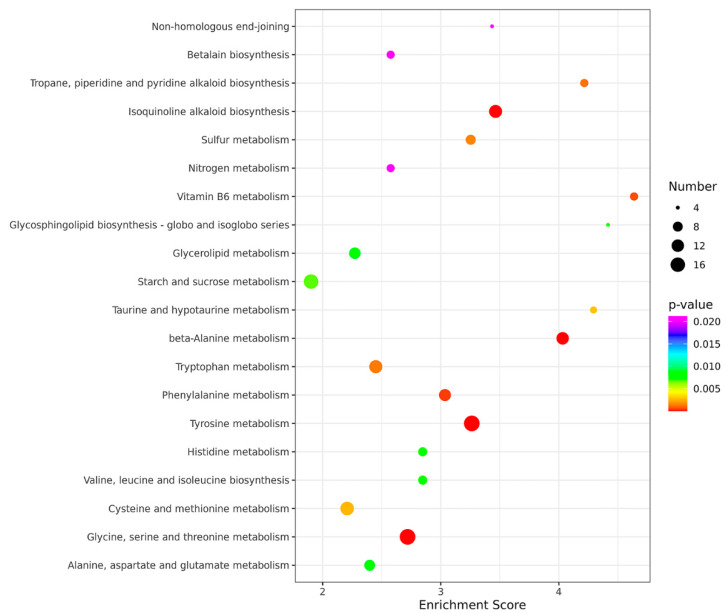
The KEGG pathways enriched in the *∆moade12* mutant. The enrichment score represents the ratio of differentially expressed genes to all annotated genes in the same pathway. Higher enrichment scores correlate with the degree of enrichment. The top 20 enriched pathways are shown.

**Table 1 jof-08-00780-t001:** Differentially expressed genes related to pathogenesis in the *Δ*moade12 mutant using |log_2_FC| > 1.5 and a *q*-value < 0.05.

Gene ID	Description	Log_2_-fold Change	Name	Reference
MGG_09134	Hypothetical protein	−8.17	*MPG1*	[32]
MGG_09100	Cutinase	−7.97	*CUT2*	[33]
MGG_11693	Hypothetical protein	−4.74	*RGS7*	[34]
MGG_12530	Histidine kinase G7	−4.61	*HIK3*	[35]
MGG_10105	Hypothetical protein	−4.19	*MHP1*	[36]
MGG_05871	Hypothetical protein	−3.11	*PTH11*	[37]
MGG_14931	Zinc finger protein 32	−2.60	*VRF1*	[38]
MGG_12865	Hypothetical protein	−2.45	*MoHOX7*	[39]
MGG_03977	Hypothetical protein	−2.40	*COS1*	[40]
MGG_13762	Multidrug resistance protein 3	−2.38	*ABC3*	[41]
MGG_09693	Hypothetical protein	−2.19	*BAS2*	[42]
MGG_00750	Cytochrome b−245 heavy chain subunit beta	−1.99	*NOX1*	[43]
MGG_05659	Fungal-specific Zn(2)-Cys(6) domain-containing transcription factor	−1.75	*CCA1*	[44]
MGG_07219	Conidial yellow pigment biosynthesis polyketide synthase	−1.71	*ALB1*	[4]
MGG_02252	Tetrahydroxynaphthalene reductase	−1.65	*BUF1*	[45]
MGG_10193	Guanine nucleotide-binding protein subunit gamma	−1.64	*MGG1*	[46]
MGG_10422	C6 transcription factor, putative	−1.64	*MoAFO1*	[47]
MGG_08850	cAMP-independent regulatory protein pac2	−1.58	*MoGTI1*	[48]
MGG_01819	Glycogen phosphorylase	−1.57	*GPH1*	[49]
MGG_06069	Endoglucanase	−1.53	*MoAA91*	[50]
MGG_01062	Hypothetical protein	1.67	*ATG8*	[49]
MGG_12655	Hypothetical protein	2.01	*AVRPi9*	[31]
MGG_01294	Hypothetical protein	2.09	*MoAGO3*	[51]
MGG_03468	Intracellular hyphae protein 1	2.34	*SLP2*	[29]
MGG_12942	Hypothetical protein	2.50	*SPD2*	[30]
MGG_10097	Intracellular hyphae protein 1	7.41	*SLP1*	[29]

## Data Availability

The data are contained within the article or in the Appendix A.

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
