# Peer review of "Adenylsuccinate Synthetase MoADE12 Plays Important Roles in the Development and Pathogenicity of the Rice Blast Fungus"

_jof, 2022, doi:10.3390/jof8080780_

Round 1
Reviewer 1 Report
This manuscript describes the forward genetic approach to identify mutant from T-DNA insertion lines and characterizes the mutant. Both knock out and complement MoADE12 mutant were generated. Pathogenicity, vegetative growth, sexual characteristics, abiotic stresses and transcriptome between WT and mutant were compared. Authors conclude that MoADE12 plays important roles in conidiation and pathogenicty of rice blast fungus.
In general, this work was carried out in a straight forward fashion. The finding is interesting and worth publishing however, the manuscript lacks critical interpretation and discussion of the results.
Some sentences in Introduction is not clear and should be re-write (Line 29-31 and 34-36). Methods should add more details so readers can repeat the experiments. For example how many rice plants were used for pathogenicity assay?
There is no method for cell wall integrity, osmosis stress and oxidative stress. Please add.
For discussion:
Line 308: need more discussion on how and why?
Line 330-334: need more discussion about genes that might be important like purine transporters.
Authors should show RNAseq data that relates to genes that involve in the discuss topics. Did authors find these transporter genes in the RNAseq data. Are they up- or down- regulated?
Authors did not discuss about KEGG result and the KEGG pathways.
Line 342-343: need more discussion on the effect of time point and the amount of ROS and the expression level of peroxidase genes.
Minor points:
Line 33: nutrients?
Line 164: fungus?
Figure 2 should include scale bar
Figure 3 A please check "H2O2" is it water H2O (This is showing that the methods do not have enough details.
Author Response
Dear reviewer,
Thank you very much for your comments and professional advice. These opinions help to improve academic rigor of our article. Based on your suggestion and request, we have made corrected modifications on the revised manuscript. We hope that our work can be improved again. Furthermore, we would like to show the details as follows:
- Some sentences in Introduction is not clear and should be re-write (Line 29-31 and 34-36). Methods should add more details so readers can repeat the experiments. For example how many rice plants were used for pathogenicity assay? There is no method for cell wall integrity, osmosis stress and oxidative stress.
Our answer:
- Line 29-31 and 34-36 had been re-write.
The details as follows:
“Endogenous resources in conidium, including glycogen, trehalose, polyols, and lipid bodies, are sufficient to support the development of infection structures in this process [5]. Once the fungus has penetrated into the first epidermal cell, the development of invasive hyphae requires the uptake of nutrients from the infected host site. During compatible interactions between M. oryzae and the plant host, the fungus derives its nutrients from living plant cells by conserved sensitive global genetic regulatory mechanisms of carbon and nitrogen metabolism before switching to the necrotrophic stage [6, 7]. These regulatory pathways, including the target of rapamycin (TOR) sig-naling cascade, carbon catabolite repression (CCR) and nitrogen metabolite repression (NMR) confer the ability to utilize a wide range of carbon and nitrogen sources, which is conserved in fungi and enable fungi to colonize diverse ecological niches, including infected host cells [6, 8].”
- Method for cell wall integrity, osmosis stress and oxidative stress has been added. Other methods also had been detailed.
The details as follows:
“2.6. Assays for osmotic stress, cell wall integrity and oxidative stress
To test the sensitivity against osmotic regulators, cell-wall-disrupting agents or oxidants, vegetative growth was assayed on CM plates with sodium chloride, sorbitol, calcofluor white (CFW, Sigma-Aldrich, St Louis, MO, USA), Congo red, rose Bengal or H2O2 (30 % (w/w) in H2O) at the different concentrations described in Results at 28°C. All reagents were purchased from Sinopharm (Shanghai, China), except for CFW. Growth rates of the tested strains were determined by measuring the colony diameter of 6-day-old cultures. All experiments were repeated at least three times. The mean ± SD of growth rate was determined using SPSS Statistics 22 (IBM, Inc., Armonk, NY, USA). Asterisks indicates a statistically significant difference at p < 0.05.”
- Line 308: need more discussion on how and why?
Our answer:
This part has been added in the revised manuscript.
The details as follows:
“In the de novo purine biosynthesis, MoADE1 is the key enzyme in the synthesis of IMP, which is the precursor of AMP and GMP. MoGUK2 catalyzes the conversion of GMP to GDP, and MoADE12 catalyzes the first committed step in the conversion of IMP to AMP. AMP and GMP can produce a variety of derivatives, some of which may mediate opposite biological responses [52]. Phenotypic differences of their gene deletion mutants may be due to the differences of their products and derivatives.”
- Line 330-334: need more discussion about genes that might be important like purine transporters. Authors should show RNAseq data that relates to genes that involve in the discuss topics. Did authors find these transporter genes in the RNAseq data. Are they up- or down- regulated? Authors did not discuss about KEGG result and the KEGG pathways.
Our answer:
This part has been added in the revised manuscript.
The details as follows:
“From our results, one putative uric acid-xanthine permease gene UAP1 (MGG_08056) were found to be downregulated in the Δmoade12 mutant grown on CM medium. Thus, the attenuated growth of infectious hyphae in the Δmoade12 mutant may be related to limited adenine availability, and the Δmoade12 mutant showed reduced growth in MM with a low concentration of adenine (Figure 1C). There is no clear indication that AMP or adenine is present in the composition of CM medium, but their analogues should be present. According to the KEGG pathway, we found that the several amino acid and carbohydrate metabolism pathways were enriched, which allow the Δmoade12 mutant to synthesize the necessary AMP in CM medium for growth.”
- Line 342-343: need more discussion on the effect of time point and the amount of ROS and the expression level of peroxidase genes.
Our answer:
This part has been added in the revised manuscript.
The details as follows:
“In the genome of M. oryzae, total 27 peroxidase genes had been revealed [56]. In the present study, transcription of 13 genes were perturbed by deletion of MoADE12. Among those, 10 genes were downregulated, including MoLIP2, CPXB, CATB, MoHPX3, MoHPX2, NOX1, MoLIP3, MoPRX1, NOX2 and MoAPX1. According to Mir et al. [56], six of those genes, including CPXB, CATB, NOX1, MoLIP3, NOX2 and MoAPX1, were upregulated under H2O2 stress, and nine of those genes, except for MoHPX3, were induced during early phase of host infection. Furthermore, CATB [33], NOX1 [43], NOX2 [43] and MoPRX1 [56] were required for pathogenicity by balancing the intracellular level of ROS or efficient removal of extracellular ROS in M. oryzae. Although deletion of CXPB does not affect pathogenicity, CXPB is involved in fungal defense against H2O2 accumulated in epidermal cells of rice at the early stage of infection [57]. Those evidences suggested that the ROS burst at the infection site is inhibitory for the growth of infectious hyphae in the Δmoade12 mutant.”
- Minor points:
5.1 Line 33: nutrients?
5.2 Line 164: fungus?
5.3 Figure 2 should include scale bar
5.4 Figure 3 A please check "H2O2" is it water H2O (This is showing that the methods do not have enough details.
Our answer:
- It has been revised.
- It has been revised.
- The scale bar has been added.
- Method has been added.
30 % hydrogen peroxide (w/w) in H2O were used to make CM medium containing H2O2 (0.01%, 0.02%, and 0.03% [w/v]) .
Thank you very much for your attention and time. Look forward to hearing from you.
Your sincerely,
Zhen Zhang
10 July, 2022
Reviewer 2 Report
Interesting paper. Sound methods. Need to clean up the writing and include numbers (statistics) to support certain statements.
Line 156 - Need to further explain the relevance of lack of growth with supplementation with hypoxanthine
Figure 1 - Use a single orientation for the figures, ie all sample identifiers above the figures and sample axis top and treatment identifiers on side
Figure 1 legend - need to define media types in the figure legend. Each figure should be stand alone.
Line 163 - change to "...no aerial hyphae, as indicated by a small amount of light colored growth on the plate (Fig 1B).
Figure 2 - Need to add size bars on photos and include that information in the legend also.
Figure 2 legend - change to (B) Perithecia (Rows 1 and 2) and asci (Row 3).
Figure 3B - Need to inlcude size markers. Also need to include arrows and label figure to show appressoria or the "Bulbous" infectious hyphae.
Need to add real numbers and statistics to prove the differences in the statements for
Line 190 - lower number
Line 181 - reduced number
Line 192 - What way were they similar? Based on what?
Line 200 - How many lesions were found?
Figure 4 legend - relative rate compared to what on which media?
Line 244 - compared with growth of Guy11 on which media?
Discussion - Delete lines 293-297 and begin with your work.
Author Response
Dear reviewer,
Thank you very much for your comments and professional advice. These opinions help to improve academic rigor of our article. Based on your suggestion and request, we have made corrected modifications on the revised manuscript. We hope that our work can be improved again. Furthermore, we would like to show the details as follows:
- Line 156 - Need to further explain the relevance of lack of growth with supplementation with hypoxanthine
Our answer:
This part has been added in the revised manuscript.
The details as follows:
“Although the process of purine salvage can generate AMP from adenine, or IMP from hypoxanthine, these results showed that AMP synthesis from IMP is disabled in M. oryzae Δmoade12 and M76.8, and the loss of MoADE12 was associated with growth defects in M76.8.”
- Figure 1 - Use a single orientation for the figures, ie all sample identifiers above the figures and sample axis top and treatment identifiers on side
Our answer:
This part has been revised. In the Figure 1C, the results of the mutnat M76.8 are added.
- Figure 1 legend - need to define media types in the figure legend. Each figure should be stand alone.
Our answer:
This part has been revised.
The details as follows:
“Figure 1. Deletion of the MoADE12 gene in M. oryzae resulted in adenine autotrophy and reduced aerial hyphae growth on nutrient-rich medium. (A) The Δmoade12 mutant was grown on GMM plates (6 cm diameter) supplemented with adenine or hypoxanthine at the indicated concentrations for six days. (B) Colony morphology was examined on the CM and OMA plates (9 cm diameter) for nine days. (C) The Δmoade12 mutant was grown on GMM plates (6 cm diameter) supplemented with adenine at the indicated concentrations for six days.”
- Line 163 - change to "...no aerial hyphae, as indicated by a small amount of light colored growth on the plate (Fig 1B).
Our answer:
This part has been revised.
- Figure 2 - Need to add size bars on photos and include that information in the legend also.
Our answer:
This part has been revised.
- Figure 2 legend - change to (B) Perithecia (Rows 1 and 2) and asci (Row 3).
Our answer:
This part has been revised.
- Figure 3B - Need to inlcude size markers. Also need to include arrows and label figure to show appressoria or the "Bulbous" infectious hyphae.
Our answer:
This part has been revised.
- Need to add real numbers and statistics to prove the differences in the statements for
8.1 Line 190 - lower number
8.2 Line 181 - reduced number
8.3 Line 192 - What way were they similar? Based on what?
8.4 Line 200 - How many lesions were found?
8.5 Figure 4 legend - relative rate compared to what on which media?
8.6 Line 244 - compared with growth of Guy11 on which media?
8.7 Discussion - Delete lines 293-297 and begin with your work.
Our answer:
This part has been revised.
8.1 From Figure 2B (row 2 and 3), the Δmoade12 mutant produced fewer perithecia than the wild type, which is visible. At that time, there was no statistics on the number of perithecia, so it was not rigorous enough that the word “lower number” was used there. Thus, “The results showed that the Δmoade12 mutant produced perithecia, but the number was lower relative to the wild type and complemented transformant (Figure 2B). Mature perithecia and asci were observed under the microscope in the Δmoade12 mutant and were similar to those in the wild type (Figure 2B).” had been revised to “The results showed that the Δmoade12 mutant produced perithecia, and the ability of the Δmoade12 mutant to produce perithecia was visually reduced compared that of the wild type and complemented transformant (Figure 2B). However, mature perithecia and asci were observed under the microscope in the Δmoade12 mutant (Figure 2B)”
8.2 The Δmoade12 mutant hardly formed conidiophores compared to the wild type. Thus, “The results showed that the Δmoade12 mutant formed substantially reduced numbers of conidiophores compared to the wild type and complemented transformants (Figure 2A)” was revised to “The results showed that the Δmoade12 mutant hardly formed conidiophores compared to the wild type and complemented transformants (Figure 2A)”.
8.3 What we mean is that the Δmoade12 mutant produced perithecia, and mature perithecia and asci were observed. Thus, “However, mature perithecia and asci were observed under the microscope in the Δmoade12 mutant and were similar to those in the wild type.” was revised to “However, mature perithecia and asci were observed under the microscope in the Δmoade12 mutant.”
8.4 We recalculated the lesions, the results showed that the lesion number of the Δmoade12 mutant was decreased 82.3% compared to the wild type. This part has been added in the revised manuscript.
8.5 Growth rate of the indicated strains exposed to different chemicals compared to that of their own on CM media.
8.6 compared with growth of Guy11 on CM media in the presence of the tested oxidants.
8.7 This part has been revised.
Thank you very much for your attention and time. Look forward to hearing from you.
Your sincerely,
Zhen Zhang
10 July, 2022
Round 2
Reviewer 1 Report
Authors have responded to all comments and suggestions. I believed the revised manuscript can be accepted for publication.
